# Multi-scale quantification and modeling of aged nanostructured silicon-based composite anodes

Thomas Vorauer[1], Praveen Kumar[2], Christopher L. Berhaut [3], Fereshteh F. Chamasemani [1], Pierre-Henri Jouneau[2], David Aradilla[3], Samuel Tardif [2], Stephanie Pouget[2], Bernd Fuchsbichler[4], Lukas Helfen[5,6], Selcuk Atalay[7], Widanalage D. Widanage[7], Stefan Koller[4], Sandrine Lyonnard[3] & Roland Brunner [1✉]

Advanced anode material designs utilizing dual phase alloy systems like $Si/FeSi_2$ nano-composites show great potential to decrease the capacity degrading and improve the cycling capability for Lithium (Li)-ion batteries. Here, we present a multi-scale characterization approach to understand the (de-)lithiation and irreversible volumetric changes of the amorphous silicon (a-Si)/crystalline iron-silicide (c-$FeSi_2$) nanoscale phase and its evolution due to cycling, as well as their impact on the proximate pore network. Scattering and 2D/3D imaging techniques are applied to probe the anode structural ageing from nm to µm length scales, after up to 300 charge-discharge cycles, and combined with modeling using the collected image data as an input. We obtain a quantified insight into the inhomogeneous lithiation of the active material induced by the morphology changes due to cycling. The electrochemical performance of Li-ion batteries does not only depend on the active material used, but also on the architecture of its proximity.

[1] Materials Center Leoben Forschung GmbH, A8700 Leoben, Austria. [2] University of Grenoble Alpes, CEA, IRIG-MEM, Grenoble 38000, France. [3] University of Grenoble Alpes, CEA, CNRS, IRIG, SyMMES, Grenoble 38000, France. [4] Varta Micro Innovation GmbH, A8010 Graz, Austria. [5] Institute for Photon Science and Synchrotron Radiation, Karlsruhe Institute of Technology, D-76131 Karlsruhe, Germany. [6] Institut Laue–Langevin, CS 20156, 38042 Grenoble Cedex 9, France. [7] WMG, University of Warwick, Coventry CV4 7AL, UK. ✉email: Roland.Brunner@mcl.at

 **1**

Silicon (Si)-based lithium-ion batteries are among the most promising candidates for decentralized storage systems in the area of renewable energy, e-mobility, or mobile electronic devices[1–7]. The main advantage to add silicon into lithium (Li)-ion anode materials is that the theoretical specific capacity of silicon ($Li_{15}Si_4$ with 3578 mAh g$^{-1}$) is about ten times that of graphite ($LiC_6$ with 372 mAh g$^{-1}$)[8–12]. Generally, in a Si-based anode material, the Si-particles are embedded in a porous graphite matrix. The porous graphite acts as a more or less mechanically robust active phase, as it undergoes only small volumetric changes of less than 10% upon lithiation and delithiation[13,14]. The ionic conductivity is primarily determined by the morphology of the porous network[15]. To ensure sufficient electrical conductivity, nanoscale conductive particles like carbon black are dispersed within a polymeric binder[16,17]. This carbon-binder domain (CBD), which accommodates the silicon, forms a network that provides electrical conductivity between the electrochemically active particles. However, silicon undergoes high volumetric expansions upon lithiation (up to 300%)[18] leading to insufficient lifetime expectancy due to degradation and high capacity fade. The volumetric expansion and contraction during cycling result in mechanically induced changes in the microstructure of the anode material such as fracture, peeling off or delamination[10,19–22]. In addition, it leads to the unfavorable continuous solid electrolyte interface (SEI) formation limiting the migration of the lithium ions between the electrolyte solvent and the active material, which finally causes irreversible capacity loss[23]. Therefore, a major challenge for Si-based anode materials is to achieve moderate electrode swelling and to control the SEI formation to avoid the capacity fade. Several strategies have been evaluated in the last years to solve this issue[24–31]. Advanced material design strategies for the anode material utilizing for instance dual-phase alloy systems have shown potential to decrease the capacity degrading and improve the cycling capability[32]. Recently, the behavior of a promising amorphous silicon (a-Si)/crystalline iron-silicide (c-FeSi$_2$) alloy nano-composite anode has been investigated[33]. It could be shown that the nanoscale organization of active and close to inactive constituents enhances cycling stability[28,34–38]. In general, a deep understanding of the structure–property relationship on all relevant lengths is in fact critical to yield design guidelines to improve the different cell component materials beyond the state of the art. However, the increased complexity of hierarchically structured anode materials makes the structural characterization and quantification on all length scales demanding. The development and application of multi-method workflows enabling more advanced characterization from μm down to atomic scales, including high-resolution 3D imaging possibilities and modeling, is crucial. This is, generally a key to unravel the behavior of complex active materials for batteries including cathode materials[39,40].

In this paper, we present a multi-scale characterization and modeling approach to collect quantified structural information at all relevant length scales, in order to analyze how the evolution of the a-Si/c-FeSi$_2$ compound particles during cycling affects Li-ion diffusion within the proximate pore network. We utilize (1) scattering techniques mainly to understand the (de-) lithiation and volumetric irreversible structural changes of the a-Si/c-FeSi$_2$ nanoscale phase, and combine those with (2) 2D/3D imaging methods to quantify the evolution of a-Si/c-FeSi$_2$ compound particles, as well as their impact on the proximate pore network with cycling numbers of up to 300 cycles. The experimental characterization is complemented by modeling the local concentration of Li-ion in the anode material, thereby providing important information about the inhomogeneous lithiation of the active material.

## Results

**Electrochemical characterization.** Figure 1 represents the cycling profiles of the a-Si/c-FeSi$_2$/graphite//Li half-cell for 100 cycles (see Supplementary Fig. 1) and the capacity retention over 300 cycles of a full 30 mAh a-Si/c-FeSi$_2$/graphite// Ni$_{1/3}$Mn$_{1/3}$Co$_{1/3}$O$_2$ (NMC) cell. In Fig. 1a, the presence of a-Si and graphite is made clear by the combination of the voltage profile of Si with the Li$_x$C$_6$ phase transition plateaus. The cycling profiles at C/10, C/5, and C/2 change very little with only 4% reversible loss of capacity when increasing the current-rate (C-rate) to C/2. A more important profile change occurs when the C-rate is increased to 2C. This is known as electrode activation polarization and is caused by a limitation of the electrode lithiation kinetics at high C-rates[41]. The high capacity retention of the composite electrode at C-rates lower and equal to C/2 is demonstrated in Fig. 1b. After 300 cycles, the cell capacity has yet to pass below 70% of the initial capacity measured at C/2 (see Supporting Information for the data obtained over 100 cycles in half-cell configuration, Supplementary Fig. 1). However, when the cell is cycled at 2C, the capacity drops fast due to polarization of the electrode. According to the results presented in the Supporting Information (Supplementary Fig. 1), it is estimated that the cell capacity at C/2 should

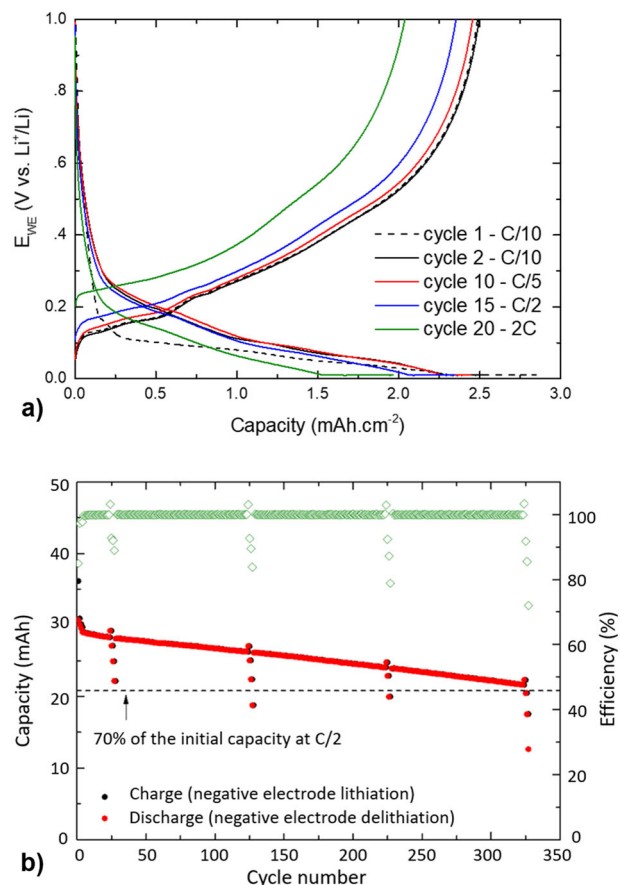

**Fig. 1 Power capability and cycling performance of the a-Si/c-FeSi$_2$/ graphite composite anode. a** Potential vs capacity of the a-Si/c-FeSi$_2$/ graphite anode measured in half-cell configuration at C/10 (black), C/5 (red), and C/2 (blue), showing only 4% reversible loss of capacity after 15 cycles. The 2C data (green) can be explained by electrode activation polarization, e.g., limiting electrode lithiation kinetics at high C-rates. **b** Capacity (black in charge and red in discharge) and efficiency (green) of a full 30 mAh a-Si/c-FeSi$_2$/graphite//NMC cell vs. cycle number. After 300 cycles, the initial capacity is still over 70% at C/2 and the efficiency stays at ~100%.

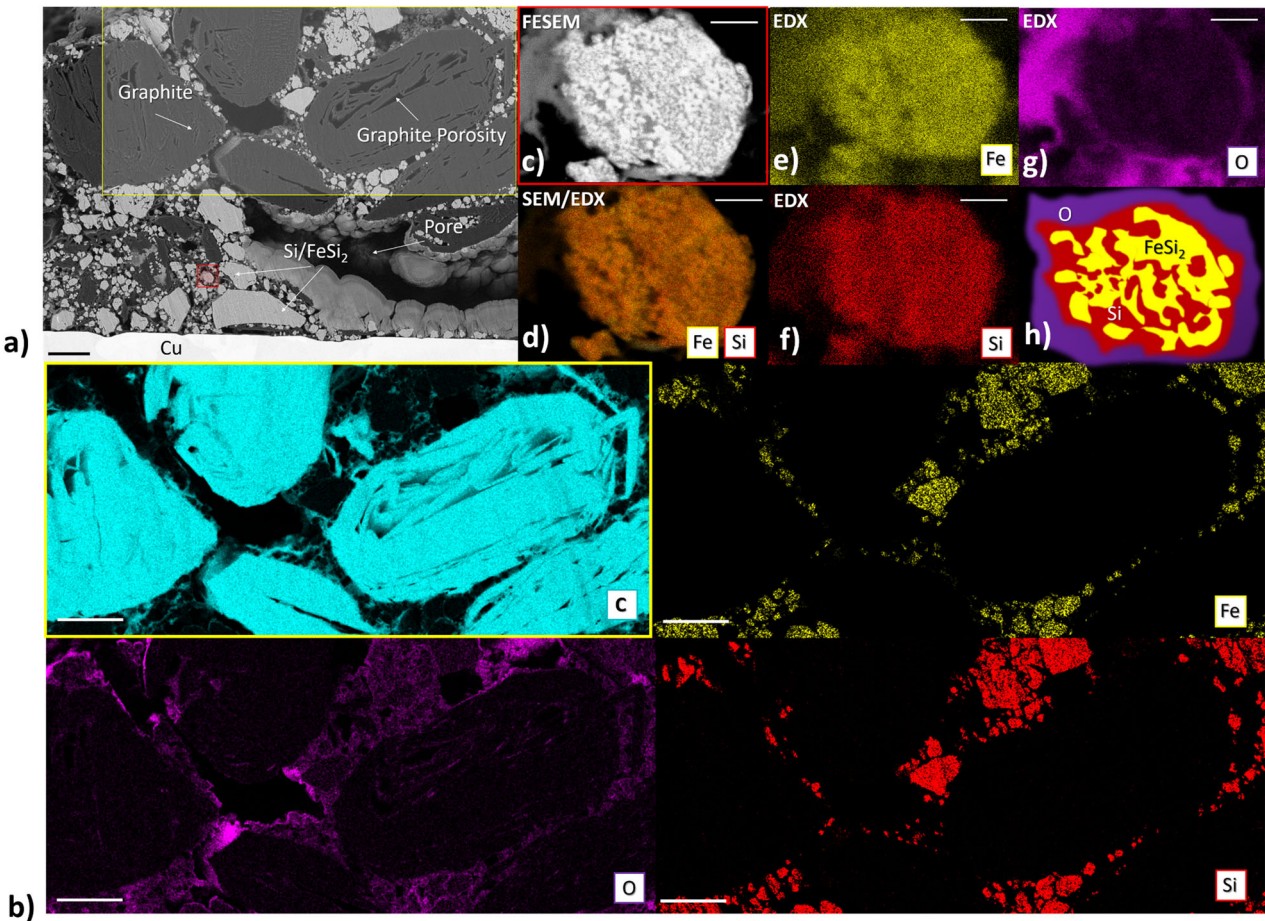

**Fig. 2 Quantification of the a-Si/c-FeSi$_2$ anode using FESEM combined with EDX. a** Backscattered electron FESEM image (ion-sliced cross-section) of the pristine anode and copper current collector. Components (graphite, a-Si/c-FeSi$_2$, pores) are labeled according to the observed gray values. The two regions (yellow and red) indicate the zoom-in for (**b**) and (**c**). **b** FESEM-EDX mapping of the yellow region for the chemical elements C (cyan), Fe (yellow), O (purple), and Si (red). **c** FESEM gray value image for a representative a-Si/c-FeSi$_2$ particle (red region) and corresponding (**d**)–(**g**) EDX mapping. **d** EDX mapping for Si and Fe, superimposed on the FESEM image, **e** Fe, **f** Si, and **g** O: no significant amount of O on the a-Si/c-FeSi$_2$ particle region. **h** Schematic showing the a-Si (red)/c-FeSi$_2$ (yellow) particle surrounded by the O (purple). The scale bar in (**a**) and (**b**) is 4 μm and in (**c**)–(**g**) 200 nm.

reach 70% of its initial value after completing at least 300 cycles following the same procedure. This is in agreement with the results obtained for a 30-mAh full-cell prepared with a NMC electrode as the cathode (Fig. 1b). The discharge capacity fading of the NMC//a-Si/c-FeSi$_2$/graphite pouch-cell is close to 30% after 321 cycles at C/2. Similar results were obtained by using a different composition of the active phases, i.e., richer in silicon[42]. These results were explained by an increase in electrode polarization caused by the negative-electrode aging mechanism and, in particular, the continuous SEI growth resulting in the loss of active lithium.

**Quantification of the anode composition on different length scales by FESEM-EDX-Mapping.** In order to gain a first insight into the morphology as well as chemical information of the composite anode on different fields of view, we combine field emission scanning electron microscopy (FESEM) together with energy-dispersive X-ray (EDX) spectroscopy. Figure 2 presents a backscattered electron FESEM image with features ranging from tens of μm down to nm. We identify in Fig. 2a according to the gray values the graphite particles (gray) with internal pores and the pore network between the graphite (black) as well as the a-Si/c-FeSi$_2$ (light gray) particles. Graphite particles and CBDs are not distinguishable by the gray value due to the similar density. However, differences in the shape of the graphite suggest a possible

differentiation. Nevertheless, we perform elemental mapping for the elements C, Si, Fe, and O on two different length scales as shown in Fig. 2b and d–g, respectively. Figure 2b illustrates that Si and Fe are intensified on the light gray particle regions. C is mainly located on the larger gray domains associated with the graphite domain but also show spots, although weaker, between the identified Fe/Si- and larger graphite domains. O is shown in particular around the Fe/Si and graphite domains and indicates a rather continuous domain. Those domains showing C and O deflection may be associated with the CBD. Figure 2c provides a zoom-in on a single a-Si/c-FeSi$_2$ particle and its vicinity. On the nm scale, clear regions with different gray value intensity (bright and darker regions) with an extension of <20 nm can be identified. The EDX mapping in Fig. 2d–f identifies those regions as Si- and Fe-rich domains, respectively. In addition, Fig. 2g provides information that those particles are surrounded by oxygen. The schematic in Fig. 2h highlights the architecture of the a-Si/c-FeSi$_2$ particles according to the experimental findings. This nanoscale organization is typical for Si/FeSi$_2$ alloy nanocomposites[33].

**Cycling behavior of the nanoscale a-Si/c-FeSi$_2$ alloy compound during lithiation and delithiation.** The behavior of the nanoscale a-Si/c-FeSi$_2$ compound structure is further investigated by scattering techniques at several states of charge (SOC). To prove that the electrode fabrication does not alter the initial morphology of

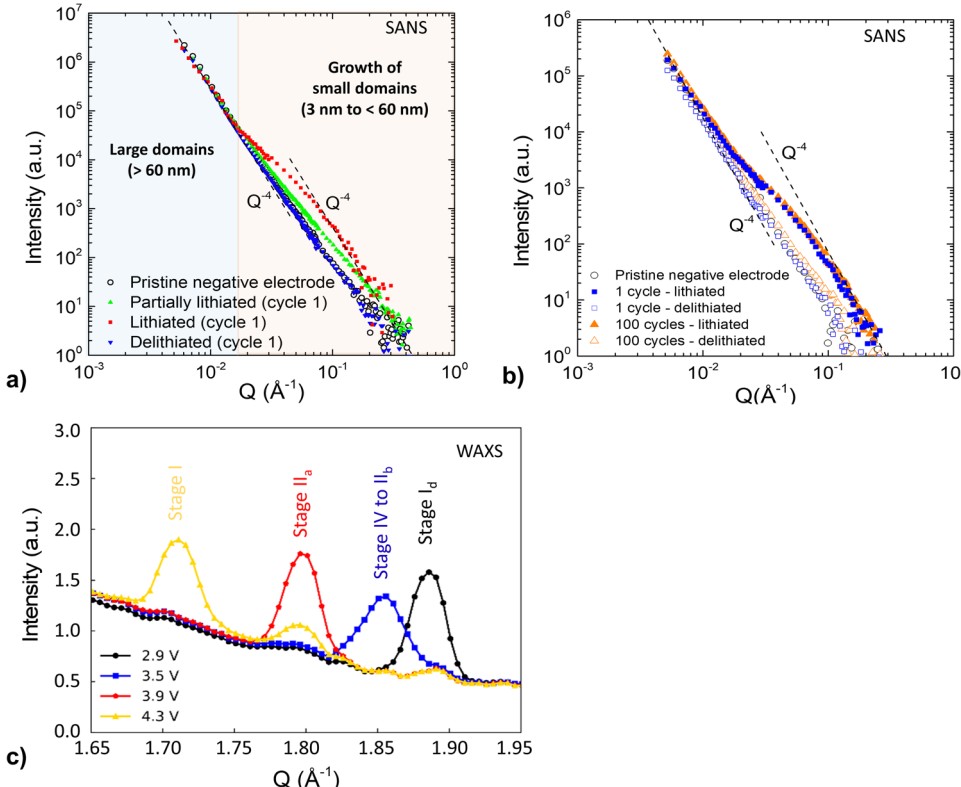

**Fig. 3 SANS and WAX results. a** SANS profiles of the negative electrode in its pristine state (black circles), at 50% (green triangles), 100% (red squares), and 0% state of charge (blue triangles). **b** SANS profiles of the (de)lithiated negative electrode after 1 and 100 cycles along with that of a pristine electrode. **c** Synchrotron WAXS data showing the $Li_xC_6$ Bragg reflections between 1.65 and 1.95 Å$^{-1}$ taken at four different potentials during lithiation of the a-Si/c-FeSi$_2$/graphite negative electrode.

the active material, we perform lab small-angle X-ray scattering (SAXS) on the Si/FeSi$_2$/graphite powder and the pristine negative electrode. Both showed similar 1D scattering profiles (Supplementary Fig. 2). Electrodes prepared at several states of charge and aging are then evaluated by post-mortem small-angle neutron scattering (SANS). The intensity versus scattering vector Q features a two-scale structure, with two separated Q$^{-4}$ decreasing laws (so-called Porod's laws) at low- and high-Q values (below $2 \times 10^{-2}$ Å$^{-1}$ and above $1 \times 10^{-1}$ Å$^{-1}$, respectively), indicating the presence of neat interfaces between scattering objects or domains of typically large and small sizes, i.e., above 60 nm and in the range of 3–50 nm, respectively. Figure 3a shows the 1D data measured during the first charge/discharge cycle at 50% (partial lithiation, green), 100% (full lithiation, red), and 0% (full delithiation, blue) SOC. Electrode lithiation produces an excess scattering intensity in the intermediate and high-Q regions, which is more pronounced after full lithiation than partial lithiation, while the low-Q Porod region is mostly maintained. We argue that the structure is modified at the scale of small domains with no significant variations of the large-scale grain interfaces. After delithiation, the electrode 1D profile is recovered, showing structural reversibility after the first cycle. Given the composition of the materials and the alloying process of silicon, the increase (decrease) in scattering intensity after lithiation (delithiation) can be attributed to the volumetric expansion (contraction) of the nanoscale a-Si phase. The aging behavior is further analyzed by comparing the 1D SANS profiles of the aged negative electrode at 0 and 100% SOC after 1 and 100 cycles (Fig. 3b). A significant excess scattering intensity is found again in the range Q = [$1 \times 10^{-2}$; $2 \times 10^{-1}$] Å$^{-1}$ for the lithiated materials after long-term cycling. At 100% SOC, the profile of the aged

sample is characterized by the same high-Q intensity level and shape, indicating that the nanoscale dimensional/morphological swelling of a-Si is not amplified from 1 to 100 cycles. After delithiation, the high-Q intensity decreases due to a-Si shrinkage but does not reverse back to the initial (pristine sample) value. Indeed, the 0% SOC SANS intensity measured after 100 cycles remains slightly higher than that measured after 1 cycle, i.e., the a-Si phase remains more expanded. During (de)lithiation, the internal nanoscale structure of a-Si/c-FeSi$_2$ particles, e.g., the defined separation between FeSi$_2$ grains and the expanded Li$_x$Si phase, is maintained, as shown in Fig. 3b.

In general, lithiation of the graphite phase is also important in Si/FeSi$_2$-graphite composites, each active component contributing to the electrode capacity according to a potential-driven sequential lithiation mechanism[42,43]. Wide-angle X-ray scattering (WAXS) is used at this stage to confirm the state of lithiation of the graphite phase in the anode. Figure 3c presents WAXS scans restricted between 1.65 and 1.95 Å$^{-1}$ to show the Bragg reflections of lithiated graphite Li$_x$C$_6$ species. The scans are taken from samples lithiated up to 2.9, 3.5, 3.9, and 4.3 V. As lithiation progresses, the sequential formation of stages I$_d$ (C$_6$ to Li$_{\approx 0.083}$C$_6$), IV (Li$_{\approx 0.167}$C$_6$), III (Li$_{\approx 0.22}$C$_6$), II$_b$ (Li$_{\approx 0.33}$C$_6$), II$_a$ (Li$_{0.5}$C$_6$), and I (LiC$_6$) can be observed by WAXS. This proves the occurrence of a Li$_x$C alloy during the lithiation process, as expected.

### 3D Quantification of the evolution of the a-Si/c-FeSi$_2$ domains and their proximity.
We may suppose that the evolution of a-Si/c-FeSi$_2$ particles has an irreversible structural impact on their proximity, e.g., on the pore network. To describe how the evolution of the a-Si/c-FeSi$_2$ affects its proximity, it is necessary to

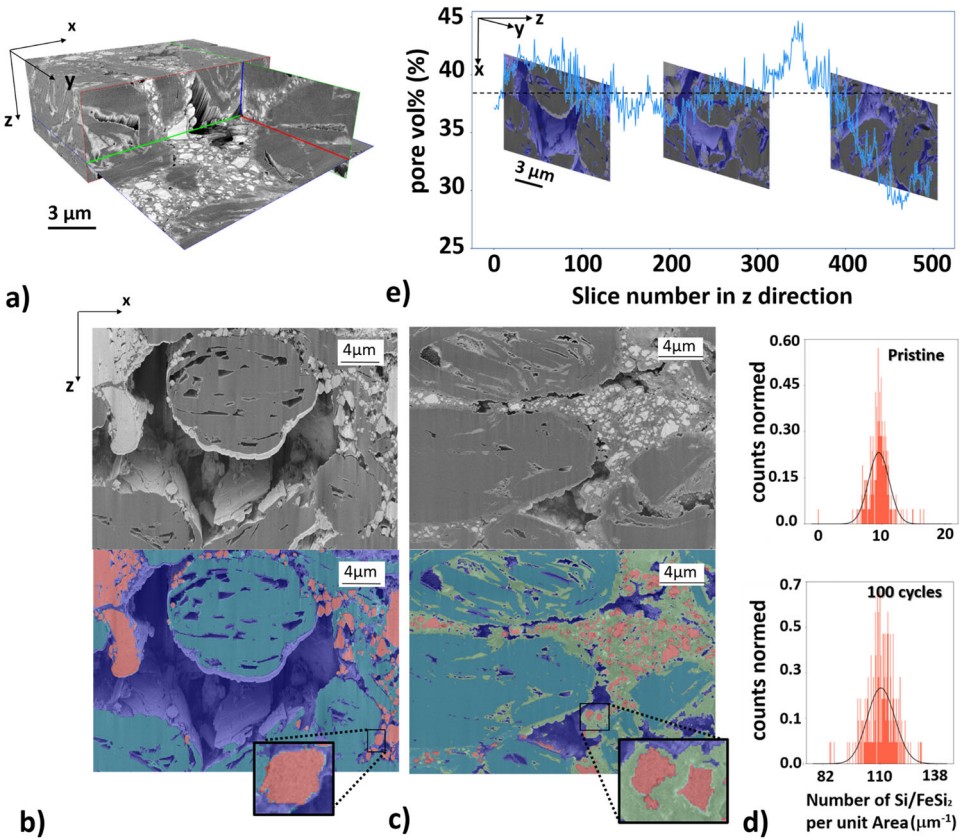

**Fig. 4 FIB-SEM tomography analysis. a** 3D reconstructed nano-FIB-SEM tomography image data of the a-Si/c-FeSi$_2$ anode, measured with the secondary electron detector (SE). The volume of interest (VOI) used for our investigations is 20 × 34 × 20 μm³ with a voxel size of 40 × 20 × 20 nm³ accordingly. Nano-FIB-SEM (SE) data for **b** the pristine anode and **c** the anode with 100 cycles. A comparison between the non-segmented (above) and segmented (below) image is presented for (**b**) and (**c**), respectively. a-Si/c-FeSi$_2$ (red), the graphite particles (cyan), and pores (blue) superimposed on the post-processed gray value data. Segmented data for the anode with 100 cycles in (**c**) shows in comparison to the pristine anode in (**b**) another phase, associated with the emerging SEI (green). Insets show a zoom-in for a representative a-Si/c-FeSi$_2$ particle. The roughness of the a-Si/c-FeSi$_2$ surface area increases with cycling. **d** Statistical distribution of the number of Si/FeSi$_2$ particles per unit area for the pristine (top) and anode with 100 cycles (bottom). **e** Statistical information of the pore vol% over about 500 slices in z-direction for the pristine anode. The maximum and mean values are 44.65% and 37.54%, respectively. The mean value is indicated by the dashed line. Three representative slice images along the z-direction are presented for further information. Pores are (blue). The scale bar in (**e**) is 3 μm for all three slice images along the z-direction.

visualize and quantify the morphology on different length scales as a function of the electrochemical cycling number as well as to estimate the Li-diffusion. In Fig. 4a, we present a typical 3D rendered image obtained from the nano-FIB-SEM tomography data. As an example, we show a representative segmented and non-segmented slice image for the pristine and 100 cycled anode material in Fig. 4b, c, respectively. For the segmentation of the FIB-SEM data, we used a combination of graph-based Felzensz-walb algorithm[44] and feature-based method (see Supplementary Note 2). Due to the shine-through artefacts we cannot segment the regions only based on their gray value. The Felzenszwalbs method[44] helps to separate the objects in each slice into about 700 separated regions. Subsequently for the identification of each region we calculate different features, e.g., average gray value, standard deviation of the gray values, number of gray values above/below a threshold. This approach allows us, in comparison to the error-prone pixel-based method, to segment the slice image data region-based and more accurately with respect to the occurring shine-through artefacts and identified domains. According to the segmentation, three different phases are identified in Fig. 4b, for the pristine anode material: the graphite (cyan), the pore network (blue), and the a-Si/c-FeSi$_2$ particles (red). Due to the similar density and gray values between the graphite particles and CBD, we do not segment the CBD from the

graphite particles. For the anode with 100 cycles (delithiated) in Fig. 4c, we identify an additional phase (green) not present in the pristine anode. This phase spreads in the proximity of the compound particles and the graphite, and is not present in the pristine anode. We argue that the due-to-cycling emerging domain (green) can be associated with the SEI growth[33]. The green domain within the graphite pores might be related to lithiated graphite Li$_x$C$_6$ species, which have been detected by the WAXS data (Fig. 3c). According to the segmented image data for the pristine material, the globular domains associated with a-Si/c-FeSi$_2$ (Fig. 4b) reveal within the corresponding region of interest (ROI) a maximum extension of up to 6.4 μm and a mean value of about 0.90 μm. In case of the pristine material, the periphery of the a-Si/c-FeSi$_2$ compound particles provides a smooth appearance. For the sample with 100 cycles, shown in Fig. 4c, the mean size in the corresponding ROI is about a factor 3 smaller (0.33 μm). A statistical analysis reveals, as illustrated in Fig. 4d for the presented ROI, that the number of particles per unit area increases significantly for the anode with 100 cycles (below) by a factor of about 10, relative to the maximum particle count in the pristine anode. The surface area of the a-Si/c-FeSi$_2$ particles now exhibits in comparison to the pristine state a rougher appearance, suggesting the formation of the tree-branch structure as reported in ref. [33]. We show in Fig. 4e that the pore volume%

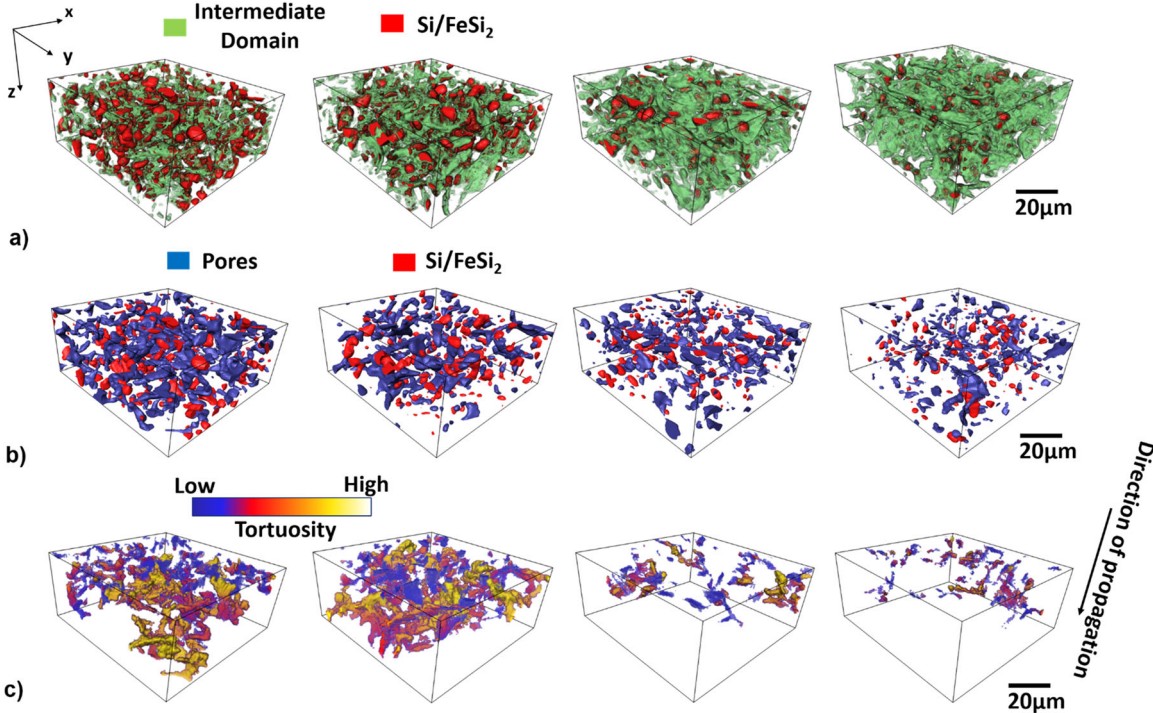

**Fig. 5 Quantified 3D-SCT data for pristine, 3, 100, and 300 (delithiated) cycles (from left to right).** As an example, for the investigations we choose a VOI with $32.5 \times 97.5 \times 97.5\ \mu m^3$ and a voxel size of $0.65 \times 0.65 \times 0.65\ \mu m^3$. **a** Segmented 3D-SCT results depict the a-Si/c-FeSi$_2$ particles (red) $\geq 0.65\ \mu m$ and the intermediate domain (green) associated with the CBD and a-Si/c-FeSi$_2$ particles with a size smaller than $0.65\ \mu m$. The green phase increases with cycling mainly due to the formation of the SEI. **b** Pores (blue) and a-Si/c-FeSi$_2$ particles (red). We can see a decrease of the average size of the pores. **c** Displays the tortuosity. Dark blue indicates a low tortuosity, i.e., only small deviations from the direct distance. Yellow indicates regions that are very hard to reach for the Li-ions coming from the surface and traveling in z-direction through the sample. White indicates infinite tortuosity, i.e., places that are not reachable using only the micro-scale pore network. The scale bar indicated on the right in (**a**)–(**c**) is 20 μm and valid for all 3D images.

for the pristine anode over 500 slices provides a mean value of 37.54% with a maximum and minimum value of 44.65% and 28.36%, respectively. Further, Fig. 4c suggests that the microscopic pore network shrinks for the sample with 100 cycles. To quantify the micro-pore network in more detail and gain more statistics, we use in the next step a volume of interest which is about 23 times larger. Therefore, we perform 3D imaging with micro-synchrotron radiation computed tomography (μ-SCT) with an effective pixel size of 650 nm and an average energy of 35 keV.

Figure 5a shows the segmented 3D-μ-SCT data for the pristine, 3, 100, and 300 cycled (delithiated) samples. We develop for the quantification of the SCT data a histogram-based global iterative threshold multistage approach to find objectively the gray value thresholds for the image data (see Supplementary Note 3 and Supplementary Fig. 4). The approach provides in the first stage the threshold for the pore- and the Si/FeSi$_2$-domain. In the next stage, we obtain an intermediate- and finally the graphite domain by using the results from the first stage and performing arithmetic operations (subtraction and addition) on the segmented image data. The intermediate domain is associated with a continuous phase surrounding the pore- and the Si/FeSi$_2$-domains. The a-Si/c-FeSi$_2$ associated regions (red) exhibit domains with extensions $\geq$ the resolution limit of $0.65\ \mu m$. We argue that, for the pristine anode, the intermediate domain incorporates the CBD as well as domains smaller than the resolution limit of the μ-SCT (Fig. 2). As shown in Fig. 5a, a significant increase (+21.8%) in the volume of this intermediate domain is observed for the anode material with 3 cycles with respect to the pristine material. The change from the pristine to the 100 and 300 cycled material is +33.3% and +38.2%, respectively. No significant change is

observed for the volume percentage of the graphite by comparing the pristine to the different cycled samples.

The strong increase of the green domain, in particular from the pristine to the material with 3 cycles may be linked to the morphological changes of the a-Si/c-FeSi$_2$ as well as to the concomitant SEI formation, as shown by nano-FIB-SEM in Fig. 4c. From these observations, one can deduce that the volume change associated with the SEI formation, as well as the redistribution of the a-Si/c-FeSi$_2$ compound particles, have direct impact on the pore network as illustrated in Fig. 5b.

The images show a change of the pore space from the pristine electrode to the material with 300 cycles. The volume of the pore network is reduced by about 37.2%, 58.8%, and 76.5% in the 3, 100, and 300 cycled electrodes relative to the pristine material, respectively (Fig. 5b).

**Spatial pore network-connectivity and Li-ion diffusion path.** To quantify the change of the 3D pore network-connectivity we calculate the tortuosity, defined as the ratio between the path through the pore network and the Euclidian distance from the surface to other points in the sample. Figure 5c shows the 3D visualization of the calculated tortuosity of the pore network for the pristine, 3, 100, and 300 cycled material (delithiated). The calculation is performed for the most relevant direction, namely from the electrode surface toward the copper current collector in z-direction. Dark blue in the color code indicates low tortuosity, meaning the ratio between the path through the pore network and the direct distance is close to 1, i.e., a very straight path. Yellow-colored areas are the regions with high tortuosity ($\gg 1$). A change of the color from blue toward yellow means that the path

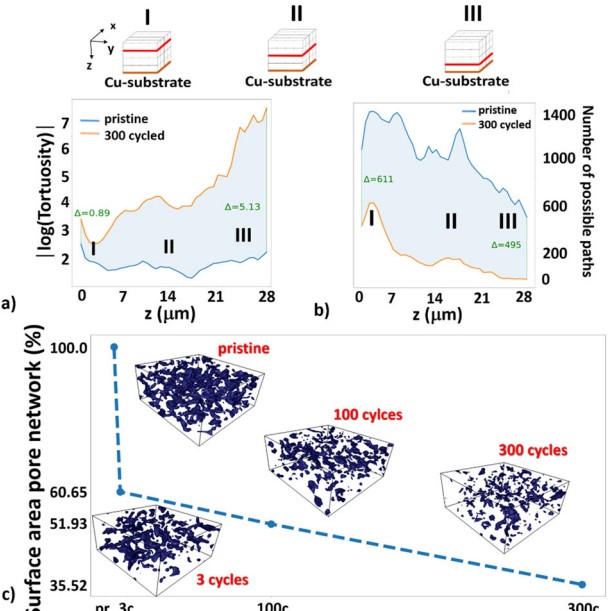

**Fig. 6 Variation of the mean tortuosity with depth z and surface area of pores. a** Compares the mean tortuosity of the pristine (dark blue line) and the 300 cycled (orange line) delithiated anode sample. The mean tortuosity difference increases significantly. The logarithm of the mean tortuosity values is plotted, as the difference between the two samples increases exponentially. I, II, and III indicate three different positions (red lines) relative to the Cu-substrate. **b** Shows the average pore connections per depth z. A decrease in the average number of connections is shown. For the sample with 300 cycles, the number of connections falls toward zero much earlier than for the pristine sample. **c** Surface area of the meshed pore network for different cycling states relative to the total surface area of the pristine sample. A steep decrease from pristine to 3 cycles is demonstrated while after 3 cycles, the dependence declines linearly. 3D images with the segmented pores (blue) are shown for pristine, 3 cycles, 100 cycles, and 300 cycles, respectively.

is getting more tortuous and the Li-ion diffusion rate decreases. White color areas correspond to infinite tortuosity, i.e., the point is not reachable from the entry side through the microscopic pore network. Clearly, we can see that the path through the pore network gets more tortuous with cycling.

In Fig. 6a, we compare the computed mean tortuosity per constant depth interval for the case of the pristine (blue line) and 300 cycles (orange line) anode. This plot shows the development from 0 up to ~28 μm depth in z-direction in logarithmic y-scale for better visualization. The mean tortuosity is increasing exponentially with the depth for the 300 cycled sample, while it is almost constant for the pristine sample. Another important parameter to quantify the diffusion through the electrode is the number of possible paths or channels per constant z, represented in Fig. 6b over the same z range. It is observed that Li-ions have more possibilities to diffuse through the pore network in the pristine material in comparison to the material with 300 cycles. At around 24 μm, only connections in the single digits are left for the diffusion in the 300 cycled material, while the pristine material still provides more than 600 possible connections. According to these data, it is therefore clear that the cycling reduces very significantly the number of available paths for lithium through the micro-pore network in the electrode, which correlates to the increased tortuosity. The decrease in channels as shown in Fig. 6b goes with a noticeable change in the surface area of the pore network, as shown in Fig. 6c. A drop of about 40% in the surface area can be observed after the first three cycles,

followed by a moderate linear-like decrease in extending cycling up to 300 cycles.

To gain more insights with respect to the Li-ion diffusion path from the pore channel to the a-Si/c-FeSi$_2$ compound particles as well as the activation performance of the Si with the Li-ions, we apply the Hausdorff metric[45]. In Fig. 7a, b, we give a 3D measure of the Hausdorff distance ($d_H$) between the pore channels and a-Si/c-FeSi$_2$ particles as well as the distributions of proximity values for the pristine and the 300 cycled sample, respectively (see also Supplementary Fig. 3 for the Hausdorff distance plots with the a-Si/c-FeSi$_2$ compound particles included). We see that the maximum (mean) distance increases significantly from 12.73 μm (3.62 μm) to 17.53 μm (5.37 μm) for the pristine and 300 cycled sample, respectively. In addition, the distribution of $d_H$ as illustrated in Fig. 7 changes with cycling. The illustrated broader distribution of the cycled anode in comparison to the pristine anode indicates an enhanced heterogeneity of $d_H$. This location dependency of $d_H$ suggests an enhanced inhomogeneous lithiation with cycling. To evaluate the impact of this structural evolution on the lithium-ion transport across the material, we proceed to a model estimation of the ion diffusion.

**Modeling the Li-diffusion in the microstructure.** The model is based on the measured, reconstructed, and image-analyzed FIB-SEM data, shown in Fig. 8a. The used ROI exhibits the pore-, a-Si/c-FeSi$_2$-, and SEI/carbon/binder (SEI/C/B) compound-domain after 100 cycles. In the model, we assume that the SEI is immobile relative to the diffusing lithium ions. Beyond the simulated diffusion kinetics, the mechanical simulation might give further details for the particle evolution for different cycling rates. A mechanical simulation is beyond the scope of this work, however, definitely an area worth further thorough investigations.

Based on the image data, our numerical approach uses a two-dimensional finite element mesh. We are using COMSOL Multiphysics® software (www.comsol.com) to define the boundaries, and mesh the domain with triangular discretization. A finer mesh is used around the surfaces of the particles and in the voids between the particles. Through systematic studies of mesh refinements, mesh independent results are established using 430k Lagrange-quadratic elements in the domain (see Supplementary Note 4 and Supplementary Fig. 6 for details). We set the pores as continuous source for the Li-ions and use a numerically evaluated area fraction (based on the ROI image) of $\varepsilon$ (0.63985) for the SEI/C/B region. We let the simulation run for 100 s. In Fig. 8a we show the modeled Li concentration projected on the image data of the aged anode (100 cycles) after 2, 10, and 100 s. Colors blue and red indicate a low and high Li concentration, respectively. Figure 8b shows the corresponding concentration profiles versus time obtained in four different areas of the ROI, labeled as I, II, III, and IV (Fig. 8a). The Li concentration reaches the steady state at different diffusion times. The time strongly depends on the area selected within the anode material. Figure 8c shows a schematic describing the change of the a-Si/c-FeSi$_2$ compound particles and its proximity on μm scale with cycling. Area I corresponds to a region close to a μ-sized pore channel. The diffusion of Li-ions is extremely fast and efficient. Here, the Li concentration changes within <10 s to 90% and indicates a high Li concentration (red) in the area as shown by Fig. 8a, respectively. In area IV the diffusion is clearly limited. The Li concentration in the vicinity of the a-Si/c-FeSi$_2$ particle is low (blue) as shown by Fig. 8a. The small a-Si/c-FeSi$_2$ particles are embedded in wide SEI/C/B regions. Here, the Li-diffusion path through the SEI/C/B compound is longer than in area I (Fig. 8c). These observations clearly point to the inhomogeneous lithiation of the active material due to their distinct environment, in

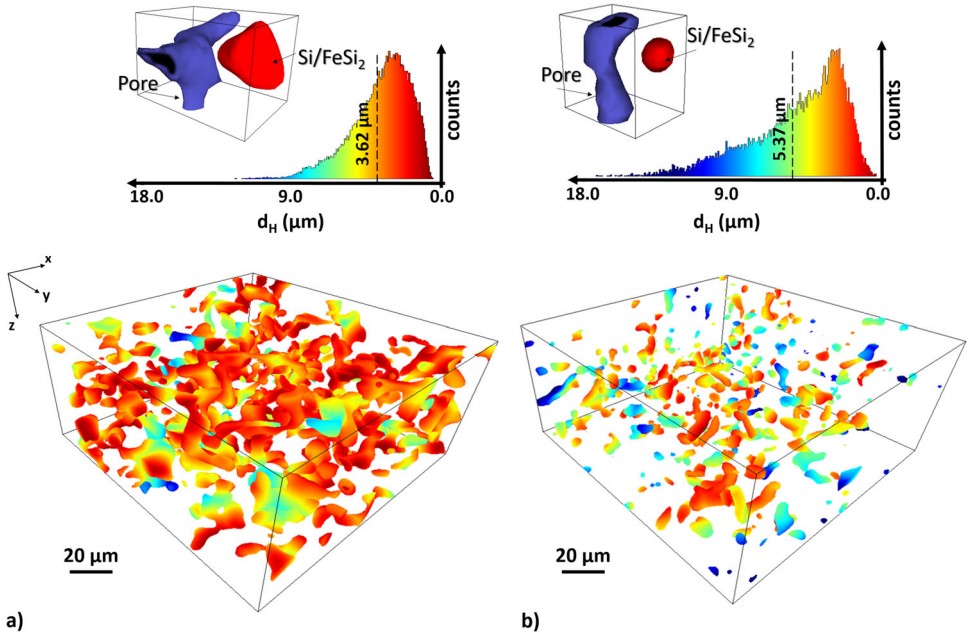

**Fig. 7 Spatial Li-ion diffusion path from the pore channel to the a-Si/c-FeSi₂ compound particles.** Hausdorff distance ($d_H$) between pore network and a-Si/c-FeSi₂ particles for the **a** pristine and **b** 300 cycled anode, respectively. Red indicates close proximity and blue indicates large distances to the next closest a-Si/c-FeSi₂ particle. The mean distance of the normed distributions of $d_H$ is getting larger from 3.62 μm in the pristine to 5.37 μm in the 300 cycled sample. The distribution (above) of $d_H$ depends on the cycling rate. The distribution for the anode with 300 cycles provides a broader distribution than the pristine anode. The 3D rendered SCT images illustrate exemplarily the different observed distances from pore (blue) to Si/FeSi₂ particle (red), for the pristine and 300 cycled anode, respectively.

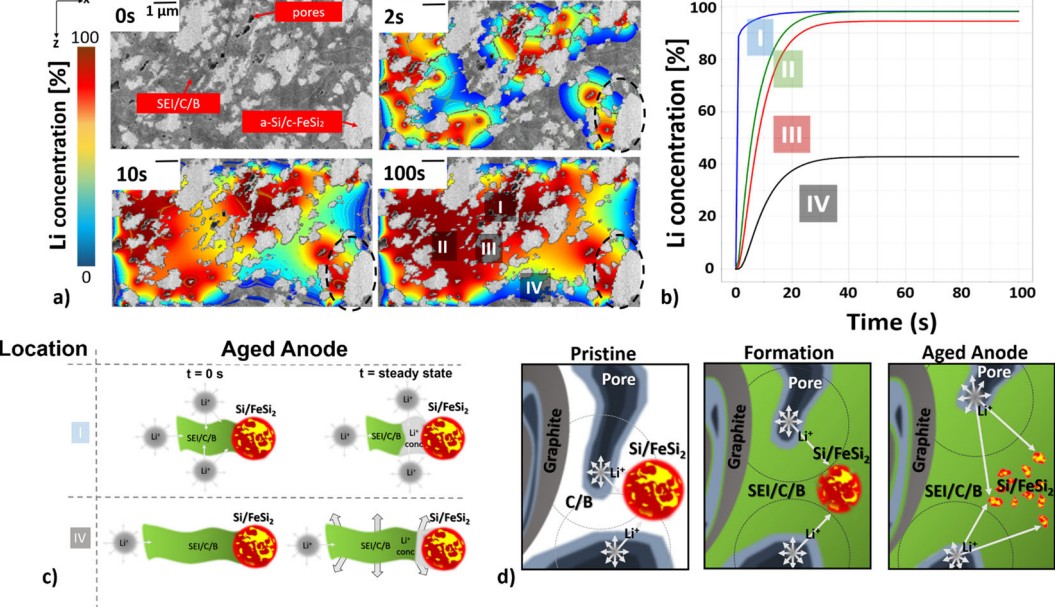

**Fig. 8 Modeling of the Li-diffusion in the aged anode after 100 cycles. a** We use a ROI of 5 × 8 μm², which shows pores, a-Si/c-FeSi₂ compound particles, and SEI/C/B domain. Modeled Li-ion concentration is superimposed on the SEM-FIB data for time step $t = 0$, 2, 10, and 100 s. Low Li concentration (blue), high concentration (red). The inhomogeneous lithiation is indicated by the colored Li-concentration profile and in particular highlighted in the areas labeled with I–IV for $t = 100$ s as well as the dashed circle indicating an a-Si/c-FeSi₂ particle for $t = 2$, 10, and 100 s. **b** The resulting Li concentration is plotted for the different regions (I–IV) as a function of time. Region I (IV) saturates fastest (slowest). **c** Schematic for position I and IV illustrates the heterogeneous lithiation for the aged anode. For region I (IV) the extension of the SEI/C/B compound is small (large). **d** Schematic of the aging mechanism on the μm scale. Pristine electrode (left): arrows indicate the distance from the pore to the a-Si/c-FeSi₂ particles. Formation step (middle): SEI formation (green), pore space decreases, a-Si/c-FeSi₂ particle morphology changes, distance increases. Aged anode (right): fractured a-Si/c-FeSi₂ particles, longest distance, and most heterogeneous distance distribution. The scale bar in (**a**) is for all images 1 μm.

particular the proximity (or not) of pores. Moreover, as can be noticed from Fig. 8a, the surface of each silicon active particle is not uniformly colored. See for instance the proximity of the large a-Si/c-SiFe$_2$ particle indicated on the lower right corner (dashed circle) in Fig. 8a (at 2 s), and how it appears after 100 s, e.g., part of its apparent surface is red (high Li-ion concentration) while other parts are blue (low Li-ion concentration). Hence, even within an individual a-Si/c-SiFe$_2$ compound particle, heterogeneous lithiation is likely to occur due to the coexistence of easily-exposed and hardly-reachable surface regions, which, again, are determined by the proximity of the Li-transporter pore network.

## Discussion

In this paper we present investigations on a-Si/c-FeSi$_2$/graphite composite anodes. As shown in ref. [33], the FeSi$_2$ alloy inclusions act as a stability factor against the mechanical forces that pull apart the silicon nano-domains due to lithiation, and help to maintain the core structure of the a-Si/c-FeSi$_2$ particles during cycling. However, there are small but irreversible changes on nm scale within the a-Si/c-FeSi$_2$ compound particles, as confirmed herein by SANS (Fig. 3b). These nanoscale structural changes may add up and trigger mechanical stress, which is not fully buffered by the architecture of the a-Si/c-FeSi$_2$ particles as well as the provided porosity. It may yield noticeable changes in the μm-scaled compound regions. Indeed, by comparing FIB-SEM results (Fig. 4) of pristine and 100 cycles (delithiated), we observe a significant increase of smaller fractured a-Si/c-FeSi$_2$ particles, showing a rougher surface, in the cycled electrode, see also the schematic in Fig. 8d.

In addition to the internal mechanical forces acting on the compounds, there are external mechanical forces coming from the growth of the SEI. Interestingly, we see that the SEI is not just surrounding the Si but grows also around and inside the large graphite particles (see green segmentation in Fig. 4c surrounding/inside the cyan graphite particles). These findings are consistent with the WAXS measurements (Fig. 3c) evidencing Li$_x$C$_6$ states.

The initial formation of the SEI and its continuous growth during cycling result in a decrease of the pore network volume (Fig. 5a, b) and surface area (Fig. 6c). The trend for the surface area (Fig. 6c) almost mirrors the behavior of the capacity curve (Fig. 1). It also shows a drastic decrease from pristine to 3 cycles, followed by an almost linear development up to 300 cycles. However, we still detect surface pores with several μm in size in electrodes cycled 500 times (Supplementary Fig. 5c). Therefore, we argue that the main skeleton of the pore network still remains. From this, we infer that the SEI growth will show a diminishing behavior (see Supplementary Note 5). This could be part of the explanation of the decrease in the slowdown of the capacity in Fig. 1. We also notice a similar gradient behavior for the increase of the SEI volume and decrease of the a-Si/c-FeSi$_2$ particle mean volume with cycling. Again, a strong change from pristine to 3 cycles, and a more moderate one to 100 and 300 cycles, is found (Fig. 5). From these results we can conclude that the greatest modifications in the negative electrode happens at the formation step, where the first SEI layers are generated (Fig. 8d). After this initial stage, the morphology and electrochemical performance of the material evolve in a more moderate way, such that the 70% capacity retention after 300 cycles can be understood.

To follow-up with the importance of pores versus SEI growth, we investigate the tortuosity (Figs. 5c and 6a, b), the Hausdorff distance ($d_H$) from the pore to the a-Si/c-FeSi$_2$ particles (Fig. 7) and simulate the Li-ion diffusion through the SEI compound to study the levels of Li-ion concentration and the time it can take for the lithium to reach the silicon reaction interface at various

regions of the aged electrode sample (see Fig. 8a, b). The modeling results for the aged anode clearly point to an inhomogeneous lithiation of the active material due to its distinct environment, in particular the proximity (or not) of pores. In particular, the results show a radial gradient for the Li concentration and it becomes clear that the closer the a-Si/c-FeSi$_2$ particles are to the pores, the faster and more evenly distributed the contact between the Li-ions and the compound particles becomes, see also the schematic in Fig. 8c. These findings are supported, by the extracted mean Hausdorff distance analysis in Fig. 7. It shows that $d_H$ increases with cycling but also shows that the distance distribution gets broader with the cycling rate. The long-term cycling results in an increased difficulty for lithium ions to reach the active sites as also indicated by the schematic in Fig. 8d. In addition, the tortuosity analysis indicates that the Li-ions have more possibilities to diffuse through the pore network in the pristine anode in comparison to the aged one (Figs. 5c and 6a, b). The broad distance distribution (see Fig. 7) may also lead to an incomplete coverage of the surface area, which may result in an uneven and unstable SEI growth. The simulation leads to the understanding that the total surface area of the a-Si/c-FeSi$_2$ particles does not necessarily take part in the (de-)lithiation process but only a specific surface area that is closest to and directed to the nearest pores does.

In conclusion, we show in this paper that the combination of scattering techniques with different 2D/3D imaging methods as well as numerical modeling provides a full understanding of the whole anode morphology with respect to (1) the volumetric irreversible structural changes of the a-Si nanoscale phase and its (de-)lithiation, (2) the evolution of a-Si/c-FeSi$_2$-, graphite-, formation of the SEI-domains and their impact on the pore network as well as (3) the inhomogeneous lithiation of the active material induced by the morphology changes, with cycling numbers of up to 300 cycles. In particular, we have shown that the electrochemical performance of Li-ion batteries does not only depend on the active material used, but also on the architecture of the proximity of the active material, like the 3D pore network. We argue that the expanded perception taking into account the whole morphology of the anode on different length scales, and its modification due to cycling, is highly crucial to yield future design guidelines and to improve the anode material beyond the state of the art.

## Methods

**Electrode preparation.** The electrode paste was produced with a *Hivis Mix Modell 2P-03/1* mixer and it was coated on a semi-industrial *Coatema Smartcoater*. The prepared electrodes were composed of 25 wt% silicon-based active material (L20772, ~1000 mAh/g), 66 wt% graphite (BTR918, ~372 mAh g$^{-2}$), 2 wt% carbon black (*Super P*) as conducting agent and 7 wt% LiPAA as binder (made from Poly Acrylic Acid MW 450,000, Sigma Aldrich), and have an area capacity of 2.40 mAh c$^{-1}$ m$^{-2}$ (more details on Supplementary Note 1). The L20772 material was provided by 3 M and contains a mixture of amorphous silicon (a-Si) with crystalline iron disilicide (c-FeSi$_2$) inclusions, and graphite. The active material present in the L20772 powder is ~20% graphite, 25% a-Si, and 55% c-FeSi$_2$ in weight.

**Coin-cell preparation.** The negative electrodes are composed of 25 wt% L20772 (a-Si and c-FeSi$_2$), 66 wt% graphite (BTR918), 7 wt% lithium polyacrylic acid (LiPAA 450) as binder, and 2 wt% carbon black (Super P), and were designed with a mass loading of 2.4 mAh cm$^{-2}$. The L20772 material was provided by 3 M and contains, in particular, silicon domains composed of finely dispersed crystalline iron disilicide (FeSi$_2$) in a continuous active amorphous silicon (a-Si) phase and graphite. The selected separator and electrolyte are, respectively, celgard 2400 (monolayer polypropylene) and 1 mol.L$^{-1}$ lithium hexafluorophosphate (LiPF$_6$) in a fluoromethyl carbonate and ethyl methyl carbonate (3FEC/7EMC, v/v) binary solvent mixture with 2 wt% vinyl carbonate (VC) as SEI forming additive. The coin-cells were assembled as 2.4 mAh cm$^{-2}$ half-cells where lithium metal is the counter electrode. Both working and counter electrodes are 10 mm in diameter.

**Electrochemical measurement.** Electrochemical measurements were performed on a MPG2 Biologic multichannel galvano-potentiostat. The cells were tested using

the same galvanostatic procedure with charge/discharge current-rates (C-rates) of C/10, C/5, C/2, and 2C. The a-Si/FeSi$_2$/graphite//Li half-cells were cycled between 1 and 0.005 V vs. Li$^+$/Li. Each charge was ended with a constant voltage step of 60 min at 0.005 V to reach maximum cell reversible capacity.

**SAXS, WAXS, and SANS sample preparation**. In total, 30-mAh pouch cells prepared using the same electrolyte and a 2.4 mAh cm$^{-2}$ negative electrode (a-Si/c-FeSi$_2$/graphite) and a 2.0 mAh cm$^{-2}$ (1/1/1) nickel–manganese–cobalt-oxide (NMC)-positive electrode were cycled 1 and 100 times to prepare negative-electrode samples. The pouch cells were then opened in an argon-filled glove box and the negative-electrode samples, taken away from the edges, sealed in aluminum laminated film pouches. Lithiated samples were prepared after 0.25 (partial lithiation), 0.5, and 99.5 cycles and the delithiated samples after 1 and 100 cycles.

**Post-mortem SANS**. The airtight Al-pouch bags were measured in transmission geometry on the D22 spectrometer at Institute Laue Langevin (ILL, Grenoble, France). Two (sample-to-detector and wavelength) configurations were used to cover the required extended range of Q. The 1D SANS profiles were obtained by radially averaging the isotropic 2D patterns. Data were corrected from sample transmission, incoming flux, and detector efficiency using routine procedures. The high-Q incoherent background due to scattering from the pouch bag and the remaining liquid electrolyte was subtracted from the electrode data.

**Post-mortem SAXS**. The airtight Al-pouch bags were measured in transmission geometry on the SAXS camera at CEA-IRIG (Cu $\lambda = 1.54$ Å). The a-Si/c-FeSi$_2$/graphite powder was measured in a standard cell with Kapton windows. The 1D SAXS profiles were obtained by radially averaging the isotropic 2D patterns. Data were corrected from sample transmission and detector efficiency using routine procedures.

**Post-mortem synchrotron WAXS**. Post-mortem WAXS measurements were performed on the BM02 (D2AM) beamline at the European Synchrotron Radiation Facility (ESRF, Grenoble, France). The energy of the incident X-ray beam was fixed at 17 keV (radiation wavelength $\lambda_{Xray} = 0.7293$ Å). WAXS powder rings were recorded using an imXPAD WOS detector. The standard sample-to-detector-distance corrections were achieved using reference materials: lanthanum hexaboride (LaB$_6$) and chromium oxide (Cr$_2$O$_3$). WAXS intensity profiles collected as a function of the momentum transfer Q were then obtained by azimuthal integration of the 2D patterns using the PyFAI library.

**FIB-SEM imaging, FESEM-EDX measurement**. 3D reconstruction of the pristine and the cycled electrodes have been performed by acquiring 2D slices of the electrode with a pixel size of 20 nm in the x and y plane while the slice thickness (z-depth spacing) was 40 nm. The final volume of the electrode was ~40 × 40 × 15 µm$^3$. A Zeiss crossbeam NVision 40 FIB-SEM was used for imaging and serial sectioning of the negative electrode simultaneously with high precision. FIB slices were cut at an accelerating voltage of 30 kV and a beam current of 700 pA with a cutting distance of 40 nm. Special care was taken for the cycled samples, all the samples were kept in an Ar filled glove box with the O and water level <1 ppm at room temperature. SEM images were acquired using secondary electron (SE) and energy selective backscattered (ESB) detectors with a primary energy of 1.5 kV and a beam current of 1 nA, to keep the lateral resolution as high as possible.

The chemical mapping of the negative anode has been performed using a field emission scanning electron microscope 450 GeminiSEM operated at 3/2KV with 3000/500 pA equipped with the Ultim Extreme energy-dispersive X-ray (EDX) system from Oxford Instruments (100 mm$^2$ detector area) and operated with window-less configuration (large ROI/small ROI). The working distance was 6/4 mm (large ROI/small ROI). For the gray value image we used the backscattered electron detector. To avoid Ga-contamination the cross-section of the anode was prepared by using a Hitachi IM400 + ion slicer.

**FIB-SEM data processing**. We carried out FIB-SEM data processing using Avizo® and Python libraries including Numpy, Scipy, Scikit-image[46] for pre-processing as described in Supplementary Note 2. For the segmentation of the FIB/SEM data, we used a combination of graph-based Felzenszwalb algorithm[44] and feature-based method (see Supplementary Note 2).

**µ-Synchrotron data measurement**. The µ-synchrotron scans were carried out at the tomography beamline ID19 of the European Synchrotron Radiation Facility (ESRF) operating at around 180 mA storage ring current with 6 GeV electron energy[47]. A pink beam from an undulator insertion device with 17.6 mm magnetic period (pre-filtered with 5.6 mm Al to isolate the second harmonic at about 35 keV average energy) was used for acquisition of 2D projection data on an indirect detector system based on a Tb-doped 4.8-µm-thick LSO scintillator, coupled via an electro-optical system[48] with ×10 optical magnification to a scientific CMOS sensor (PCO edge 5.5, PCO, Kehlheim, Germany). Per CT scan, 2999 projections at an effective pixel size of 0.65 µm were collected with 50 ms exposure time each,

resulting in a scan time of about 5.5 min. Tomographic reconstruction software developed in-house was used for on-line experiment monitoring[49] and the final reconstructed 3D images[50].

**µ-Synchrotron data processing**. We carried out image processing of synchrotron data using Python libraries (similar to FIB-SEM data). For the µ-SCT data, we developed a histogram-based global iterative threshold multistage approach to find objectively the gray value thresholds for the image data (see Supplementary Note 3 and Supplementary Fig. 4).

**3D visualization and rendering**. We used Avizo® (version 2019.1) for rendering and 3D visualization.

**3D mesh processing**. 3D triangular µ-synchrotron meshes are provided using Avizo® and processed in MeshLab (open source).

**Simulation**. The simulation was done with COMSOL (www.comsol.com). Physical parameters used in the simulations are $D_{Li} = 3.01 \times 10^{-10}$ m$^2$ s$^{-1}$ in the bulk for the Li-ion diffusion coefficient[51], the bulk concentration[51] C$_{Li}$, is 1 M, and the mass transfer coefficient $k_f = 0.001$ m s$^{-1}$. This value is estimated based on a battery system where the ion-flux within the domain is governed with surface chemistry and external applied electric field equivalent to 1C-rate and. A supplementary note (see Supplementary Note 4 and Supplementary Fig. 6) is furthermore added demonstrating the validation of the modeling approach.

## Data availability

All data that support the findings of this study are available from the corresponding author upon reasonable request.

## Code availability

All code that support the findings of this study are available from the corresponding author upon reasonable request. The simulation was done with COMSOL (www.comsol.com).

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

## Acknowledgements

We acknowledge the financial support from the European Union (EU) under the Horizon 2020 research and innovation program (grant agreement No. 685716 "Sintbat" and No. 875514 "ECO²LIB"). ESRF and ILL are acknowledged for beam time allocation and access. L. Porcar is thanked for his help on neutron experiment, N. Boudet and N. Blanc for their help on D2AM experiment at ESRF. The WOS detector used for the WAXS measurements was funded by the French National Research Agency (ANR) under the "Investissements d'avenir" program with the grant number: ANR-11-EQPX-0010. The electrochemical work has been performed with the use of the Hybriden facility at CEA-Grenoble (France). We acknowledge support from B. Sartory and J. Wosik, both MCL, for the FESEM-EDX measurements.

## Author contributions

R.B., T.V., and F.F.C. planned and performed the image analysis work. S.A. and W.D.W. performed the simulations. S.K. and B.F. fabricated and provided the samples. L.H. and R.B. performed the μ-synchrotron measurements. R.B. planned and conducted the FESEM-EDX measurements. S.L., D.A., C.L.B., S.P., and S.T. planned and performed the SANS, WAXS, and SAXS measurements. C.L.B. performed the electrochemical measurements. P.-H.J. and P.K. planned and conducted the SEM/FIB and STEM-EDX (Supplementary Fig. 5) measurements. All authors discussed the results and commented on the paper. T.V. and R.B. wrote the paper with discussions mainly with S.L., C.L.B., P.K., and W.D.W.

## Competing interests

The authors declare no competing interests.
