## [Peer Review File · Communications Chemistry]

Reviewers' comments:

Reviewer #1 (Remarks to the Author):

In this manuscript, the authors presented a multi-scale characterization approach to understand the (de-)lithiation and irreversible volumetric changes of the a-Si/c-FeSi₂ nanoscale phase and the evolution of the a-Si/c-FeSi₂ particles, as well as their impact on the proximate pore network. It was found that the electrochemical performance of Li-ion batteries depend not only on the active materials used, but also on the architecture of the proximity of the active material. Overall, the design of the experiment is reasonable, and well planned. I would like to recommend it for publication in Communications Chemistry after major revision as follows:

(1) The author labeled multiple components including the graphite, carbon/binder, and the Si/FeSi₂ in Figure 2a, how do you distinguish these components? The SEM EDX mapping results over a larger field of view should be provided. Besides, the scale bar and the shape of the Si/FeSi₂ particle in Figure 2b are not match with the marked region in Figure 2a.

(2) The author mentioned that a combination of graph based and feature based method (Supplementary Note 2) for the FIB/SEM data is used for the phase identification. The detailed information about using these features to perform the segmentation should be provided, especially identify the SEI phase.

(3) Why does the author choose the red rectangle area for the calculation of the particle density per unit area? There are different regions contain the particle in Figure 4b. The statistical significance shall be addressed.

(4) The author mentioned that the a-Si/c-FeSi₂ compounds and the graphite contribute to 50.5 % and 49.5 % of the total capacity, respectively. The details in Supplementary Note 4 is misleading, because based on your calculation method, these two values are not correlated with the total areal capacity (2.4 mAh/cm²).

(5) There are some mistakes in the manuscript, e.g. "Figure (SF2)" at Page 5, the "Figure (4b)" at Page 8, "Li-Ion" at Page 21.

Reviewer #2 (Remarks to the Author):

The work by Vorauer provides quantitative characterization of the silicon based composite electrodes. The author applied multiple techniques to evaluate how the microstructure transforms after cycling. Overall, the work is interesting but needs revisions. The reviewer has a few comments:

1. There has been similar work on the cathode, such as layered oxide cathodes, for example, <https://doi.org/10.1038/s41467-020-16233-5>, and <https://doi.org/10.1002/aenm.201900674>. Can the authors comment on the similarities and differences between the methods presented here and in the literature?

2. The tortuosity analysis is interesting. It would be great if the authors can also present the evolution of the tortuosity at different SOCs.

3. The mechanical aspect of the analysis is not quite discussed in the manuscript. Can the authors provide some discussion?

4. Can the authors provide experimental evidence to establish the correlation in this statement? cited from abstract "...thereby providing important information about the inhomogeneous lithiation of the active material." Can the degree of lithiation be spatially resolved as a function of local Li diffusion?

Reviewer #3 (Remarks to the Author):

The article "Multi-scale quantification and modeling of aged nanostructured silicon-based composite anodes" provides detailed description of the FIB-SEM and X-ray imaging methods used for characterization of a-Si/c-FeSi₂ materials. The paper is structured well and it is written in good English. I think it can be accepted by Communications Chemistry after minor revisions.

Nevertheless, there are several possibilities for improvements which should be considered:

1. We can recognize the pore and Si/FeSi₂ from Figure 2a, but it is difficult to distinguish graphite and carbon/binder. The author should provide evidence that what is carbon/binder, rather than others. It is better to give a more detailed information or add more discussions.
2. Why is there such a big pore in the pristine electrode in Figure 2a? is the tap density influenced? How does it influence the performance? Please provide the figure of SEM to determine the particle size of Si and FeSi₂. I think it will make your readers better understand the big pores here.
3. FeSi₂ material is purchased, please provide its phase structure such as XRD. Is there any additional information we can obtain about FeSi₂ material?
4. Please provide energy range of synchrotron computed tomography in this work. In Figure 4 and Figure 5, for the 3D reconstructed data, how does the author distinguish SEI, carbon, binder and pores, please give the corresponding explanation.
5. In paper, "The strong increase of the green ("additional") phase....." (line 242) and ".....additional phase can be observed....." (line 237). Please explain to the specific meaning of "additional phase" in text.
6. The image quality is not good enough. Please provide a high quality image to facilitate readers' reading and understanding.
7. Scale label is fuzzy, such as Figure 3a and 3b. I would suggest the authors to give a check all the scale label and modify it.

Point-by-point response to Referees 'comments:

Please see our detailed response point-by-point for each referee below. The associated main changes in the text are highlighted by yellow in the revised manuscript.

Referee 1:

In this manuscript, the authors presented a multi-scale characterization approach to understand the (de-)lithiation and irreversible volumetric changes of the a-Si/c-FeSi₂ nanoscale phase and the evolution of the a-Si/c-FeSi₂ particles, as well as their impact on the proximate pore network. It was found that the electrochemical performance of Li-ion batteries depend not only on the active materials used, but also on the architecture of the proximity of the active material. Overall, the design of the experiment is reasonable, and well planned. I would like to recommend it for publication in Communications Chemistry after major revision as follows:

We thank Reviewer 1 for his/her questions and remarks. In the following, we will address each question:

- Q1)

The author labeled multiple components including the graphite, carbon/binder, and the Si/FeSi₂ in Figure 2a, how do you distinguish these components? The SEM EDX mapping results over a larger field of view should be provided. Besides, the scale bar and the shape of the Si/FeSi₂ particle in Figure 2b are not match with the marked region in Figure 2a.

Thank you for the comment. We addressed the comment and think that the manuscript improved significantly by adding additional chemical analysis for different field of views. According to the referees comment we completely modified Figure (2) by adding FESEM-EDX mapping for the pristine electrode for two different field of views to identify the different components on μm and nm scale.

Based on the grey value or threshold we are able to distinguish between the pores (dark grey or black), graphite (grey) and Si/FeSi₂ domains (light grey). Based on the grey value selection approach binder and graphite are hardly to distinguish due to the similar density and similar grey values. However, to distinguish between the two components one may take in additionally the characteristic shape as well as the distribution under consideration. Graphite especially when not calendered gives rather large ellipsoidal particles contrary to continued regions. Continued regions in the anode usually can be mainly associated with carbon binder domains.

Nevertheless, for the present paper in order to avoid any confusion we indicate in the modified version of the grey value FESEM image (Figure (2)) just the graphite, a-Si/c-FeSi₂ and pore domains based on the grey value approach. To gain further chemical information as suggested by the Reviewer, we now present FESEM-EDX mapping for the chemical elements C, Fe, Si and O for different fields of view. The new Figure (2) illustrates that both elements Fe and Si are concentrated on the regions showing the light grey domains in the grey value image, presumably associated with the a-Si/c-FeSi₂ domains. C is mainly located on the larger grey domains (graphite) but also show spots, although weaker, between the identified a-Si/c-FeSi₂ - and larger graphite domains. O is shown in particular around the a-Si/c-FeSi₂ domains

as well as between the graphite and α -Si/c-FeSi₂ domains and indicates a rather continuous domain. We argue that according to the EDX analysis the regions between the graphite regions and α -Si/c-FeSi₂ particles showing C and O can be associated with the carbon/binder domain. This is in accordance with the information submitted in the previous manuscript. We modified the figure caption and text in the manuscript accordingly. Scale bar and marked regions are now matching.

Main changes in manuscript on page 6 and 7, "Quantification of the anode composition on different length scales by FESEM-EDX-Mapping": Figure 2, Figure caption and Text.

- Q2)

The author mentioned that a combination of graph based and feature based method (Supplementary Note 2) for the FIB/SEM data is used for the phase identification. The detailed information about using these features to perform the segmentation should be provided, especially identify the SEI phase.

We added according to the referee's comment additional information in the manuscript to explain the used method in more detail. We tried to make the text clearer with respect to the reviewers comment. Due to the shine-through artefacts, we cannot use simple thresholding methods to segment the different phases. Therefore, we applied Felzenszwalb's algorithm to find the boundaries of the different regions. From that, we receive about ~ 700 separated regions per slice image. To identify each of those regions, we calculate a number of features like the mean grey value and standard deviation, to classify each region as α -Si/c-FeSi₂, graphite or background/pore. This approach also helps us to identify accurately the different phases (pore, graphite and Si/FeSi₂) for the pristine anode as well as the emerging additional phase for the anode with 100 cycles, which is not present in the pristine anode. We argue that this emerging phase based on the grey value can be associated with the grown SEI. The average grey value for the SEI regions is in-between graphite and Si/FeSi₂.

Main changes in manuscript on page 11, 12 and 13, "3D Quantification of the evolution of the α -Si/c-FeSi₂ domains and their proximity": Figure 4, Figure caption and Text.

- Q3)

Why does the author choose the red rectangle area for the calculation of the particle density per unit area? There are different regions contain the particle in Figure 4b. The statistical significance shall be addressed.

We modified Figure 4 according to the reviewers comment. We addressed the statistical significance and included a statistical comparison of the number of Si/FeSi₂ particles per unit area as a function of the cycling rate for the whole ROI illustrated in Figure (4) and modified the text accordingly. The data taken of the whole region of interest shows that the amount of particles per unit area changes significantly from pristine to the anode with 100 cycles.

Main changes in manuscript on page 13, "3D Quantification of the evolution of the α -Si/c-FeSi₂ domains and their proximity": Figure 4, Figure caption and Text.

- Q4)
The author mentioned that the a-Si/c-FeSi₂ compounds and the graphite contribute to 50.5 % and 49.5 % of the total capacity, respectively. The details in Supplementary Note 4 is misleading, because based on your calculation method, these two values are not correlated with the total areal capacity (2.4 mAh/cm²).

Our estimation of the contribution is based on the assumption that only the active components namely the graphite and the Si-powder L20772 are mainly relevant for the calculation of the total capacity of 2.4 mAh/cm². Nevertheless, to avoid any confusion we deleted the sentence in the manuscript as well as deleted the section in the Supplementary Note 4.

- Q5)
There are some mistakes in the manuscript, e.g. "Figure (SF2)" at Page 5, the "Figure (4b)" at Page 8, "Li-Ion" at Page 21.

Thank you for pointing those mistakes out. We corrected the mistakes.

Referee 2:

The work by Vorauer provides quantitative characterization of the silicon based composite electrodes. The author applied multiple techniques to evaluate how the microstructure transforms after cycling. Overall, the work is interesting but needs revisions. The reviewer has a few comments:

We want to thank the 2nd Referee for his/her questions:

- Q1)
There has been similar work on the cathode, such as layered oxide cathodes, for example, <https://doi.org/10.1038/s41467-020-16233-5>, and <https://doi.org/10.1002/aenm.201900674>. Can the authors comment on the similarities and differences between the methods presented here and in the literature?

We appreciate very much that the referee brought forward this very interesting papers on the characterization of NMC based cathode materials. We cited now both very interesting papers in the state of the art, indicated as reference 39 and 40.

In common of the mentioned work and our work is definitely, that an in-depth understanding of the role that the electrode microstructure plays in modulating the performances, is necessary.

However, a significant difference in the mentioned two works and our work are the components of interest. The cathode and the anode definitely display two major components of the cell, however show different materials, morphology and functionality and therefore need different approaches for the characterization. In our work we focus on the anode, in particular on a nanostructured Si-based composite anode.

In the following, we tried to summarize the main points of those works and oppose those to our work. We assign the work presented in <https://doi.org/10.1038/s41467-020-16233-5> with (A) and our work with (B).

1.)

(A) **deals with** the characterization of a **thin cathode** with a monolayer of **active NMC particles up to 10 cycles**.

(B) deals with the characterization of **nanostructured silicon (Si/FeSi₂ compound) –based hierarchically structured composite anode** relevant for industrial applications **up to 300 cycles**.

2.)

(A) **visualizes three main phases** namely NMC particles + carbon binder domain (CBD) + pores using phase contrast tomography at the charged state.

(B) **visualizes three (up to four when cycled) main phases** namely Graphite + Si/FeSi₂ nanocomposites + Pores + (SEI) using synchrotron micro-tomography and nano FIB/SEM tomography.

3.)

(A) focus on the relatively large features caused by the **NMC particle detachment** from the CBD network.

(B) focus on the multi-scale characterization on the **nanostructured silicon-based composite anode** to understand (1) the (de-)lithiation and irreversible volumetric changes of the nanoscale Si, (2) the evolution of α -Si/c-FeSi₂-, graphite-, SEI-domains and their impact on the pore network as well as (3) the inhomogeneous lithiation of the active material induced by the morphology changes, with cycling numbers of up to 300 cycles.

4.)

(A) **does not investigate the fine pore structure** within the CBD.

(B) investigates in particular the **pore structure (pore network)**, its change due to different cycling configurations as well as the **heterogeneity of the lithiation** from the Si/FeSi₂ particles (Spatial tortuosity analysis and distance analysis based on the 3D morphology data in combination with simulation based on experimentally determined morphology information). In particular, we show **that the electrochemical performance** of Li-ion batteries does not only depend on the active material used, **but also on the architecture of the proximity of the active material**, like the 3D pore network.

5.)

(A) develops a numerical modeling **to calculate the spatial distribution of the electrical resistance** over the surface of the NMC particles based on the phase contrast tomography data.

(B) develops a numerical modeling to obtain **insight into the location dependent diffusion rates of the Li-ions** within the heterogeneous anode materials based on the tomographic data, thereby providing important information about the **inhomogeneous and location dependent lithiation** of the active material. Discussion in combination with experimentally determined and analyzed 3D morphology data.

6.)

(A) develops a **machine-learning model** due to statistical requirements (automatic processing of **650 NMC particles**) and the lack of labeling with conventional algorithm (watershed and separation algorithm).

(B) uses the **graph based Felzenszwalb algorithm** for the nano-SEM/FIB data and a developed **histogram/based iterative global threshold multistage approach** to segment the different phases and quantify the morphological modification of the pore network, nano-Si/FeSi₂ compound domain and SEI. **High statistical** output for the current investigations on the anode material are **not necessary**, therefore a machine learning based algorithm is not necessary. The **non-machine-learning based algorithm provides good accuracy**.

7.)

(A) uses complementary **X-ray phase contrast imaging correlated with spectroscopic imaging to analyze the NMC modification** for the cathode, since the NMC's electron density is a fundamental physical property that changes upon charging and discharging.

(B) **applies complementary small angle neutron scattering** to study the morphological modification of the nanostructured Si/FeSi₂-compound as well as, **wide angle scattering** to investigate the modification of the graphite phase upon charging and discharging.

In the following, we tried to summarize the main points of those works and oppose those to our work. We assign the work presented in <https://doi.org/10.1002/aenm.201900674> with (A) and our work with (B).

1.)

(A) deals with the characterization of **polycrystalline nickel-rich layered NMC composite cathode material up to 50 cycles**.

(B) deals with the characterization of **nanostructured silicon (Si/FeSi₂ compound) –based hierarchically structured composite anode** relevant for industrial applications **up to 300 cycles**.

2.)

- (A) Combines hard X-ray phase contrast nano-tomography (cathode particle's structural degradation at electrode level), nanoscale hard X-ray spectroscopy (microcracks and chemical response in a charged NMC with up to 50 cycles), soft X-ray absorption spectroscopy (XAS: chemical implication of different degree of structural degradation), transmission electron microscopy (TEM: cathode particle's structural degradation) to systematically investigate the morphological and chemical degradation in polycrystalline nickel-rich layered NMC composite cathode material under fast charging conditions.
- (B) Combines small angle neutron scattering (Si-nano domain modification with cycling), small angle X-ray (Si- powder characterization) and wide angle x-ray scattering (graphite modification with cycling), nano-FIB-SEM tomography (3D morphology modification with respect to Si/FeSi₂-domains and particle statistics, SEI formation and porosity on nm scales), FESEM-EDX (chemical characterization of the anode on nm and μm -scales), synchrotron micro-computed tomography (3D morphology characterization with respect to SEI formation and 3D pore-network degradation up to 300 cycles and impact on Li diffusion and activation on micro- scales) and electro-chemical RVE based simulations to support the experimental findings and provide further information about the heterogeneity of the lithiation process from the Si/FeSi₂ particles.

3.)

- (A) quantification of the **degree of local damage** in the **composite cathode** and quantify the depth-dependent trend of particle fracturing within the cathode.
- (B) focuses on the multi-scale characterization of the nanostructured silicon-based **composite anode** to understand **(1)** the (de-)lithiation and irreversible volumetric changes of the nanoscale Si, **(2)** the evolution of α -Si/c-FeSi₂-, graphite-, SEI-domains and their impact on the pore network as well as **(3)** the inhomogeneous lithiation of the active material induced by the morphology changes, with cycling numbers of up to 300 cycles.

4.)

- (A) uses finite element modeling to provide insight into the **cathode strain distribution** and evolution based on RVE collected by the nanotomography to understand the depth heterogeneous electrochemistry and mechanics of the NMC composite electrode.
- (B) develops a numerical modeling to obtain insight **into the location dependent diffusion rates** of the Li-ions within the cycled anode material based on RVE-tomographic data and understand the importance of taking the whole morphology of the anode into account to yield future design guidelines.

5.)

- (A) investigates **cracking patterns depth profiles of 10-cycled and 50-cycled cathodes**

(B) 3D pore network and Si/c-FeSi₂ modification from pristine- 3, 100 and 300 cycled anodes.

- Q2)

The tortuosity analysis is interesting. It would be great if the authors can also present the evolution of the tortuosity at different SOCs.

Very interesting point. For this publication, we performed SCT measurements for the de-lithiated samples. A proposal for further SCT measurements with a lithiated/de-lithiated anode material in this context has been submitted earlier this year. Due to the current worldwide situation we hope to start with the experiments by next year and hopefully may provide further interesting results in this respect.

- Q3)

The mechanical aspect of the analysis is not quite discussed in the manuscript. Can the authors provide some discussion?

With respect to the mechanical aspect, we modified the text in particular the "Discussion". The neutron small angle scattering (Figure (3)), provides information with respect to irreversible changes of the Si/FeSi₂ particles on nm-scale. We argue that these nanoscale structural changes may add up and trigger mechanical stress, which is not completely buffered by the architecture of the α -Si/c-FeSi₂ particles as well as the provided porosity. It may yield noticeable changes in the μ m-scaled compound regions. Indeed, by comparing SEM/FIB results (Figure (4)) of pristine and 100 cycles (de-lithiated), we observe a significant increase of smaller fractured α -Si/c-FeSi₂ compound particles, in the cycled anode; showing a rougher surface, see also the schematic in Figure (8d).

Mechanical simulation are definitely of interest in particular to study the evolution of the Si/FeSi₂ particles for different cycling rates, however are beyond the scope of this work. We add a sentence in this respect in the text (page 22).

Main changes in manuscript on page 22 and from 23 and 24, "Modeling the Li-diffusion in the microstructure" and "Discussion": Fig.4, Fig.8, Text.

- Q4)

Can the authors provide experimental evidence to establish the correlation in this statement? cited from abstract "...thereby providing important information about the inhomogeneous lithiation of the active material." Can the degree of lithiation be spatially resolved as a function of local Li diffusion?

Thank you for asking this question. We think that our point was not clear enough. In this respect, we modified the manuscript according to the referees question. The degree of lithiation at present form is resolved by the investigation of the morphology and its change due to the cycling for instance by the tortuosity analysis as well as the analysis via the

Hausdorff distance in combination with the performed simulations. The experimental investigations are summarized within the new chapter "Spatial pore network-connectivity and Li-ion diffusion path". We tried to make this clearer in the text and modified the Figure illustrating the Hausdorff distance and distance distribution (Figure (7)). In particular, the analysis with the Hausdorff distance shows that the mean distance between the pores and the Si/FeSi₂ domains significantly changes (Figure (7)). This analysis also provides information about inhomogeneous lithiation by looking on the distribution of the Hausdorff distance. The distribution is getting broader with cycling which suggests that a wider range of distances from min to max are present in comparison to the pristine where the distribution is smaller and a more homogeneous lithiation may take place. Further, the qualitative tortuosity analysis provides 3D information regarding the spatial Li diffusion through the anode. We modified Figure (5) in this respect. Applying the quantitative analysis based on the experimental obtained 3D image data, we gain further information regarding the diffusion of the Li-ions through the pore network for the pristine as well as cycled anode. Clearly, the possible path shrink with cycling (Figure (6)). In addition, the morphology of the Si/FeSi₂ domains changes as shown in the SEM/FIB data as well as in the SCT data. The simulation based on the representative volume element data in Figure (8) supports the argument.

Main changes in manuscript on page 16 - 20, "Spatial pore network-connectivity and Li-ion diffusion path" as well as in "Discussion" on page 25: Figure 8, Figure caption and Text.

Referee 3:

The article "Multi-scale quantification and modeling of aged nanostructured silicon-based composite anodes" provides detailed description of the FIB-SEM and X-ray imaging methods used for characterization of a-Si/c-FeSi₂ materials. The paper is structured well and it is written in good English. I think it can be accepted by Communications Chemistry after minor revisions.

We would thank the Referee 3 for his/her questions and respond to them here:

- Q1)

We can recognize the pore and Si/FeSi₂ from Figure 2a, but it is difficult to distinguish graphite and carbon/binder. The author should provide evidence that what is carbon/binder, rather than others. It is better to give a more detailed information or add more discussions.

We modified the text accordingly. We completely agree with the reviewer that it is difficult to distinguish between the graphite and carbon/binder from the observed grey values. It seems that the text written in this context as well as Figure (2) was confusing. We tried to distinguish in Figure (2) between the carbon/binder and graphite not by the grey value but rather by the shape and distribution. Graphite especially when not calendered gives large ellipsoidal particles rather than continued regions. Continued regions in the anode usually can be mainly associated with carbon binder domains.

However, we completely modified Figure (2) by providing additional experimental data for the pristine electrode for two different fields of views to quantify the different domains on μm and nm scale.

We now present FESEM EDX mapping for the chemical elements C, Fe, Si and O for different fields of view. The new Figure (2) illustrates that both elements Fe and Si are concentrated on the regions showing the light grey domains in the grey value image, presumably identified as Si/SiFe₂ particle. C is mainly located on the larger grey domains but also show spots, although weaker, between the identified Fe/Si- and larger graphite domains. O is shown in particular around the Fe/Si domains as well as between the graphite and Fe/Si domains and indicates a rather continuous domain. We argue that according to the EDX analysis the regions between the graphite regions and Si/FeSi₂ particles showing C and O can be associated with the carbon/binder domain. This is in accordance with the information submitted in the previous manuscript. We modified the figure caption and text in the manuscript accordingly. To avoid any confusion we deleted the allocation of the carbon/binder domain in Figure(2a) and present the FESEM-EDX data instead.

Main changes in manuscript on page 6 and 7, “Quantification of the anode composition on different length scales by FESEM-EDX-Mapping”: Figure 2, Figure caption and Text.

- Q2)

Why is there such a big pore in the pristine electrode in Figure 2a? is the tap density influenced? How does it influence the performance? Please provide the figure of SEM to determine the particle size of Si and FeSi₂. I think it will make your readers better understand the big pores here.

According to the comment of the reviewer, we modified Figure (4), which provides now the SEM image showing the particles as well as added additional information. We now show in Figure (4e) the behavior of the pore vol% over about 500 slices to illustrate the min (28.36 %) and max (44.65 %) as well as the mean value (37.54 %). The intention to use such a porosity was to buffer the Si volume expansion and linked mechanical stress behavior. Therefore, we used a non-calendared anode. In addition, we wanted to investigate and understand in this context the undisturbed system. Nevertheless, as indicated in Figure (2) nanoscale structural changes are observed. Those may add up and trigger mechanical stress, which is not completely buffered by the architecture of the α -Si/c-FeSi₂ particles as well as the provided porosity. Indeed, by comparing SEM/FIB results (Figure (4)) of pristine and 100 cycles (delithiated), we observe a significant increase of smaller fractured α -Si/c-FeSi₂ particles, by about a factor of ten, showing a rougher surface area, in the cycled electrode, see also the modified schematic in Figure (8d).

Figure (4) shows the representative slice image associated with the particle size analysis.

Figure (2c) shows a representative FESEM image from a Si/FeSi₂ particle. Further, chemical analysis for this compound particle is presented in Figure (2d-g). A schematic of the compound particle to provide a better understanding of its architecture is now shown in Figure (2h).

Main changes in manuscript on page 11 and 12, "3D Quantification of the evolution of the α -Si/c-FeSi₂ domains and their proximity": Figure 4, Figure caption and Text.

- Q3)

FeSi₂ material is purchased, please provide its phase structure such as XRD. Is there any additional information we can obtain about FeSi₂ material?

[REDACTED]

The FeSi₂ material is not purchased as a single material, but mixed with silicon in the composite active material powder purchased from 3M. We provide additional chemical information and the architecture of the Si/FeSi₂ material in Figure (1). The schematic based on the experimental data in Figure (2h) illustrates the morphology of Si/FeSi₂. XRD data taken in-house reveals the presence of FeSi₂ crystalline phases (mainly P4/mmm tetragonal phase with a minority orthorhombic phase). [REDACTED]

- Q4)

Please provide energy range of synchrotron computed tomography in this work. In Figure 4 and Figure 5, for the 3D reconstructed data, how does the author distinguish SEI, carbon, binder and pores, please give the corresponding explanation.

The energy was given in the Methods section (35keV 650nm). According to the reviewers comment we also put it now in the results.

Due to the limited resolution, we cannot separate the SEI, carbon and binder directly. Therefore, we develop a segmentation method based on a histogram-based global iterative threshold multistage approach to find objectively the grey value threshold of the image data. The approach provides in the first stage the threshold for the pore- and the Si/FeSi₂-domain. In the next stage we obtain a so-called intermediate- (previously additional phase) and finally the graphite-domain by using the results from the first stage and performing arithmetic operations (subtraction and addition) on the image data. Due to the iteratively obtained threshold, we obtain an intermediate domain between the graphite, pore and Si/FeSi₂ domain. The intermediate domain is associated with a continuous phase surrounding the pore- and the Si/FeSi₂ -domains in the image data. A more detailed discussion about the

approach is found in the supplementary information. We also modified the results section as well as the supplementary to make the approach clearer for the reader.

We think that the wording “additional phase” was confusing since we also used the same wording for the explanation of the SEM/FIB data. For the latter the additional domain emerged due to the cycling and is only present in the cycled electrode and not in the pristine anode, associated with the emerging SEI growth.

The intermediate domain shows for the pristine anode a certain vol% in the SCT-data and significantly grows with cycling. The vol% of Si/FeSi₂ particles and the pores decreases. We argue that due to the significant growth of the intermediate domain from pristine to 3 cycles and up to 300 cycles, it must be related to the SEI formation. This argument is also supported by the observed morphological changes seen from the SEM/FIB investigations on smaller length scales. The new modified Figure (5) also illustrates in more detail the interplay between the intermediate phase associated with the SEI and the pore domain.

Main changes in manuscript on page 13-15, “3D Quantification of the evolution of the a-Si/c-FeSi₂ domains and their proximity”: Figure 5, Figure caption and Text.

- Q5)

In paper, “The strong increase of the green (“additional”) phase……” (line 242) and “……additional phase can be observed……” (line 237). Please explain to the specific meaning of “additional phase” in text.

We modified the text according to your question. Please see also the answer to Q4 in this respect.

We also think that the wording “additional phase” was confusing since we also used the same wording for the explanation of the SEM/FIB data. For the latter the additional domain emerged due to the cycling and is only present in the cycled electrode and not in the pristine anode. We modified the text accordingly and changed in the present text the word “the additional phase” to “intermediate phase” to make the meaning more clear.

Main changes in manuscript on page 14 and 15, “3D Quantification of the evolution of the a-Si/c-FeSi₂ domains and their proximity”: Figure 5, Figure caption and Text.

- Q6)

The image quality is not good enough. Please provide a high quality image to facilitate readers' reading and understanding.

Thank you for pointing this out. We modified the images to facilitate the reader’s reading and understanding and enhanced the quality. Therefore, we also rearranged the Figures accordingly.

- Q7)

Scale label is fuzzy, such as Figure 3a and 3b. I would suggest the authors to give a check all the scale label and modify it.

We increased the size of the labels to make them better readable. Please see the new figures according to your recommendations.

REVIEWERS' COMMENTS:

Reviewer #1 (Remarks to the Author):

The authors have sufficiently addressed my comments. I recommend the publication of this manuscript in its present form.

Reviewer #2 (Remarks to the Author):

The authors have adequately addressed the comments.

Reviewer #3 (Remarks to the Author):

The concerns have been addressed after the revision,so I think it can be accepted by the Communications Chemistry.